# An Effective Deep Learning-Based Architecture for Prediction of N⁷-Methylguanosine Sites in Health Systems

Muhammad Tahir [1], Maqsood Hayat [1,*], Rahim Khan [1] and Kil To Chong [2,3,*]

1 Department of Computer Science, Abdul Wali Khan University Mardan, Mardan 23200, Pakistan; muhammdtahir@awkum.edu.pk (M.T.); rahimkhan@awkum.edu.pk (R.K.)
2 Department of Electronics and Information Engineering, Jeonbuk National University, Jeonju 54896, Korea
3 Advanced Electronics and Information Research Center, Chonbuk National University, Jeonju 54896, Korea
* Correspondence: m.hayat@awkum.edu.pk (M.H.); kitchong@jbnu.ac.kr (K.T.C.)

**Abstract:** N⁷-methylguanosine (m7G) is one of the most important epigenetic modifications found in rRNA, mRNA, and tRNA, and performs a promising role in gene expression regulation. Owing to its significance, well-equipped traditional laboratory-based techniques have been performed for the identification of N⁷-methylguanosine (m7G). Consequently, these approaches were found to be time-consuming and cost-ineffective. To move on from these traditional approaches to predict N⁷-methylguanosine sites with high precision, the concept of artificial intelligence has been adopted. In this study, an intelligent computational model called N⁷-methylguanosine-Long short-term memory (m7G-LSTM) is introduced for the prediction of N⁷-methylguanosine sites. One-hot encoding and word2vec feature schemes are used to express the biological sequences while the LSTM and CNN algorithms have been employed for classification. The proposed "m7G-LSTM" model obtained an accuracy value of 95.95%, a specificity value of 95.94%, a sensitivity value of 95.97%, and Matthew's correlation coefficient (MCC) value of 0.919. The proposed predictive m7G-LSTM model has significantly achieved better outcomes than previous models in terms of all evaluation parameters. The proposed m7G-LSTM computational system aims to support the drug industry and help researchers in the fields of bioinformatics to enhance innovation for the prediction of the behavior of N⁷-methylguanosine sites.

**Keywords:** deep learning; pattern recognition; LSTM; RNA; natural language processing

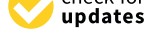

## 1. Introduction

To date, about 150 various types of RNA alteration/modification have been recognized. These changes in RNA perform important functions in regulating the expression of genes at different levels. For example, it was previously confirmed that changes in RNA can affect RNA transport, processing, mRNA translation, and stability [1,2]. The most abundant RNA modifications are the N⁷-methylguanosine (m⁷G), which happens in the 5′ cap position of mRNA molecules and transfer RNA (tRNA) loop eukaryotic S ribosomal RNA (rRNA); these modifications are preserved amid the three different kingdoms. This modification plays a precarious role in the regulation of RNA function, post-transcription modifications, and metabolism [3]. Within the context of aging, Zago et al. have discussed the emerging importance of microRNAs as biomarkers for Parkinson's disease [4]. Unfortunately, data on the functional mechanisms are very limited. Recently, it has been revealed that m7G sites can be efficiently identified by modern sequencing techniques [5–7]. Using deep sequencing technology, Marchand et al. investigated AlkAniline-Seq for identifying m7G in RNA at single-nucleotide resolution in yeast, human, and bacterial mitochondrial and cytoplasmic rRNAs and tRNAs [5]. Furthermore, differently modifying internal m7G sites to certain basic sites, Zhang et al. effectively established a MeRIP-seq approach to predict m7G sites at single-base resolution. The single-base resolution approach used in m7G-seq

data revealed the profile of m7G in human mRNA and tRNA, to enhance the knowledge of m7G distribution in human cells [6].

However, the transcriptome-wide dissemination and vibrant regulation of m7G within internal mRNA areas are still unknown. According to Zhao et al., the internal mRNA m7G methyltransferase METTL1, and not WDR4, is a key responder to post-ischemic insults, resulting in a global reduction in m7G methylation inside mRNA [8]. In addition, Liu et al. introduced m7GPredictor for predicting internal m7G modification sites using sequence properties [9]. In this model, the authors used various numerical descriptor methods and a random forest was used for the selection of optimal feature sets. Likewise, Bi et al., developed a computational model for the identification of m7G sites [10]. In this model, they have used different types of sequence encoding schemes in combination with the XGBoost algorithm. Further, Shoombuatong et al. proposed a new predictor known as THRONE for discrimination of human RNA N7-methylguanosine sites [11]. The THRONE was designed in three steps using an ensemble learning predictor. Likewise, the m7G-DPP web predictor was introduced by Zou and Yin by using physicochemical properties of RNA for the prediction of m7G sites [12]. Here, Pearson correlation coefficient, dynamic time warping, and distance correlation were utilized for extracting numerical features. Next, the LASSO algorithm was employed to select highly discriminative features [12]. Likewise, Zhang et al. introduced a predictor, namely BERT-m7G, by utilizing in staking ensemble approach for the identification of RNA m7G sites [13]. In this model, a BERT-based multilingual model was utilized to represent the information RNA sequences. Similarly, to specifically detect the internal mRNA m7G mutation, Malbec et al. developed the m7G individual nucleotide-resolution cross-linking and immunoprecipitation with sequencing (miCLIP-seq) approach [7]. Finally, this group of researchers determined that m7G modifications are enriched in AG-rich contexts, which are highly preserved in different mouse tissues and human cell lines. However, the advanced sequencing techniques revealed significant findings in this area, although these methods are still costly for transcriptome-wide detection. In this context, computational analysis m7G site predictors have been introduced, namely m7GFinder [14], iRNA-m7G [15], and m7G-IFL [16]. In these predictors, Yang et al. introduced a computational m7GFinder tool that can predict m7G sites in H. sapiens RNA using a sequence-based approach. The optimal feature subset was determined using mRMR, F-score, and Relief; and a support vector machine (SVM) was used as a classifier. Similarly, in sequential, the iRNA-m7G model was performed by Chen et al. for the identification of m7-methylguanosine sites by fusing multiple feature spaces. In this model, sequential- and structural-based features were integrated in order to form a hybrid space. Three types of features were combined using the feature fusion method, including secondary structure components, pseudo-nucleotide composition, and nucleotide property and frequency, to extract important RNA sequence features. Experiments have shown that the feature fusion technique outperforms the use of a single type of feature in detecting m7G sites [15]. Similarly, Ning et al. presented a predictor for the identification of m7G, namely m7G-DLSTM based on an LSTM model and natural language processing (NLP), nucleotide chemical property, and binary code feature extraction methods [17]. Most recently, an m7G-IFL computational model for identifying m7G sites was developed by Dai et al. [18]. This model uses an RNA sequence-encoding iterative feature representation approach to discover probabilistic distribution information from various sequential models and improve feature representation skills in a supervised iterative manner. The m7G-IFL predictor used various feature extraction techniques such as ring-function-hydrogen properties (RFH), physical-chemical-properties (PCP), and binary k-mer frequency (BKF). Then, extreme gradient boosting (XGBoost) was applied as a classifier. Furthermore, it was discovered that the proposed iterative feature method can improve feature representation capability during the iterative phase through feature analysis [16].

Furthermore, enhanced efficiency of existing computational models is still needed in the detection process. Thus, there is a dire need for the development of novel computational methods for the accurate, fast, and precise detection of m7G modification. In our recent



study, we tried to focus on deep learning-based prediction methodologies to develop an accurate computational system called "m7G-LSTM" to predict N7-methylguanosine sites, which could directly determine m7G sites based on sequence information. The proposed m7G-LSTM system contains two stages i.e., distributed attributes representation and long short-term memory (LSTM) model. In the attributes representation stage, the NLP-based approach word2vec is applied to fragment the RNA instance into words (3-mers). Likewise, in the second stage, the N7-methylguanosine site is identified by using the LSTM model. The proposed prediction model m7G-LSTM has shown better performance and obtained promising outcomes.

## 2. Methods and Materials

### 2.1. Benchmark Dataset

Here, we select and download a benchmark dataset from Chen et al. [15,16] to train the proposed computational system. The benchmark dataset consists of 741 positive sequences, which are m7G sites, and 741 negative sequences, which are non-m7G sites; both have the same length (41 nucleotides). The benchmark dataset is mathematically expressed in Equation (1).

$$S = S^+ \cup S^- \tag{1}$$

The benchmark dataset $S$ consists of m7G sites and non-m7G sites, $S^+$ with positive m7G sites sequences, and $S^-$ with negative non-m7G sites sequences. To examine the performance of the proposed model, cross-validation can be used. The dataset is split into three sections: 20% for testing, 10% for validation, and 70% for training.

### 2.2. Encoding Scheme

The one-hot encoding approach is a simple but useful feature extraction technique, frequently used in deep learning, but shows effective performance in bioinformatics [19] and computer science [20]. It is employed to illustrate the nucleotide acid composition along the RNA/DNA sequence. In previous studies [21–25], one-hot encoding was employed. In this encoding technique 'A', 'C', 'G', and 'U' are represented by binary vectors of (1, 0, 0, 0), (0, 1, 0, 0), (0, 0, 1, 0), and (0, 0, 0, 1), respectively. As a result, an n nucleotide RNA/DNA sequence is encoded as a $4 \times n$ dimensional binary vector, which is used as input to the CNN and LSTM models in this study. The vector has a length of $n = 41$ nucleotides. Figure 1 demonstrates the graphical representation of the one-hot encoding scheme.

| | A | C | C | G | U | U | G | A | C |
|---|---|---|---|---|---|---|---|---|---|
| A | 1 | 0 | 0 | 0 | 0 | 0 | 0 | 1 | 0 |
| C | 0 | 1 | 1 | 0 | 0 | 0 | 0 | 0 | 1 |
| G | 0 | 0 | 0 | 1 | 0 | 0 | 1 | 0 | 0 |
| U | 0 | 0 | 0 | 0 | 1 | 1 | 0 | 0 | 0 |

**Figure 1.** Description of the one-hot encoding scheme.

### 2.3. Distributed Attributes Formulation

The distributed attributes formulation scheme reduces the classification error of the computational model by obtaining noiseless data. Genetic data are usually expressed as biological sequences; hence, it may be thought of as a language through which information moves between cells. Natural language processing (NLP) has been used for a variety of biological problems in this area, such as EP2vec [26], alternative splicing site [27], G2Vec [28], and iN6-Methyl (5-step) [29]. We approach this sequence analysis problem from a new angle, manipulated by NLP. Indeed, there are several effective deep learning applications in the NLP, i.e., word2vec, which embeds words into a vector space. The paragraph vector is built on word2vec, and it embeds whole phrases into vectors that encode their semantic content. In this regard, treating the sequence of RNA/DNA as a sentence rather than

an image is more natural since DNA sequences are just one-dimensional data, whereas images are frequently two-dimensional data. Consequently, we consider a DNA/RNA sequence to be a sentence made up of k-mers (or words) [26]. Here, an NLP-based method, i.e., word2vec, is applied to obtain decipherable demonstrations for RNA sequences. For discontinuity, the RNA sequences are first fragmented into multiple words represented by overlapping k-mers. Here, the value of k = 3 indicates a 3-mer. Commonly, genomes are collected from the Genbank databank by using the following link: http://hgdownload. soe.ucsc.edu. The genome is split into distant 21 chromosomes (C1, C2, C3, C4, C5, . . . , C20, and C21). Additionally, the chromosome is fragmented with sentences of 100 nt residues. Lastly, the words are created by cutting each sentence into overlapping 3-mers. The word2vec model is trained using the continuous bag-of-words (CBOW) technique. The current word w(t) is predicted using the context words around it in a predetermined frame in the CBOW technique. Table 1 shows the training parameters of the word2vec model. Finally, each 3-mer word is expressed by a 100-dimensional vector, and each sequence of length L is represented by an array of shapes (L − 2) × 100.

**Table 1.** Training parameters of word2vec.

| List of Parameters | Word2vec Model |
|---|---|
| Training Method | CBOW |
| Corpus | Human Genome |
| Context words | 3-mer |
| Vector size | 100 |
| Window size | 5 |
| Minimum Count | 5 |
| Negative Sampling | 5 |
| Epochs | 20 |

*2.4. Convolutional Neural Networks (CNN)*

A CNN is a deep learning algorithm frequently used in image processing, natural language processing, and bioinformatics studies [30–37]. In image data, CNN works with two-dimensional; however, CNN can also be employed with three-dimensional and one-dimensional data. In this regard, the 1D (1-dimensional) CNN model in the field of bioinformatics is effectively applied [24,38–41]. A CNN comprises one input layer, multiple hidden layers such as pooling layers, ReLU (activation function) layers, convolutional layers, normalization layers, fully connected layers, and an output layer. In this study, various optimal hyper-parameters, such as the size of the masks [3, 5, 7, 9, 11, 13, and 15], the number of masks [4, 6, 8, 10, 12, 14, and 16], and convolution layers [1, 2, and 3] are used for training CNN model. The dropout probability range was [0.2, 0.25, 0.3, 0.35, 0.4, 0.5, 0.6, 0.7, and 0.75]. The selections of hyper-parameters are performed on the best success rates in terms of all performance metrics to discriminate N7-methylguanosine sites. The convolution layer, ReLU layer, and sigmoid function are mathematical as follows:

$$\text{Conv1D}(R)_{jf} = \text{ReLU}\left(\sum_{s=0}^{S-1}\sum_{n=0}^{N-1} W_{sn}^{f} R_{j+s,n}\right) \tag{2}$$

In Equation (2), $R$, $f$, and $j$ stand for the input, filter index, and output index position, respectively. $N$ shows the number of input channels, and $S$ denotes the size of the window.

Dense layer with dropout: The scalar output score of the dense layer is transformed from the feature vector $z$.

$$f = w_{d+1} + \sum_{k=1}^{d} w_k z_k \tag{3}$$

$$f = w_{d+1} + \sum_{k=1}^{d} m_k w_k z_k \tag{4}$$

In Equations (3) and (4), $w_{d+1}$ represents the term of additive bias and the previous layer $z_k$ weight is $w_k$. The rectified linear function is denoted by ReLU and mathematically stated in Equation (5).

$$\text{ReLU}(z) = \max(0, z) \tag{5}$$

As its output is scaled to the [0, 1], the sigmoid function is responsible for predicting whether a given sequence is an m7G site or not. Equation (6) expresses the sigmoid function mathematically.

$$\text{Sigmoid}(z) = \frac{1}{1 + e^{-z}} \tag{6}$$

### 2.5. Long Short-Term Memory Layer (LSTM)

The recurrent neural network (RNN) is a type of deep learning that can learn only from sequential data such as time-series data and textual data [42]. However, it has the issue of gradient vanishing, and thus the parameters are not updated during the backprop-agation [36,43–46]. Therefore, LSTM is a type of RNN that may store information regarding long-distance data dependence and added gating function by addressing the issue of RNN gradient [30,47,48]. LSTM gating mechanisms enable the network to effectively decide to keep it remember or ignore it. Furthermore, speech recognition and language translation also have great contributions [49–51]. Figure 2 illustrates our proposed model, which is composed of an input layer, two LSTM layers, and followed by a dense layer.

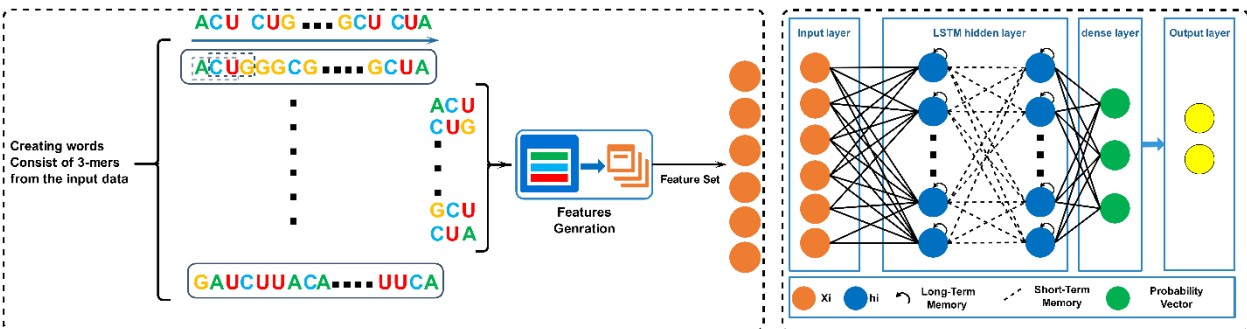

**Figure 2.** The proposed m7G-LSTM computational model Schema.

The first LSTM layer has an output channel of size 32, which is fed into the second LSTM layer, where the second layer has 64 output channels. Moreover, the dropout rate of 35% is applied to the input connection within the LSTM layers. The outcome of the second LSTM layer is flattened and passes to the dense layer. The dense layer is a fully-connected layer with x output channels, and it is followed by a sigmoid activation function. Finally, the sigmoid function generates the outcomes.

The proposed model has been trained as follows. Let $x_i$ be a vector demonstrating the input RNA sequence (Equation (7)). The LSTM computes $z_i$ for $x_i$ (Equation (9)). Sigmoid (Equation (10)) changes $z_i$ to a vector of values between 0 and 1. The loss is the binary cross-entropy of the prediction (Equation (10)). It is used for updating the hidden neurons at the hidden layer utilizing the Adam optimization algorithm, with 0.0005 being set as the learning rate.

$$x_i = RNA \text{ sequence where } x_i \in \{A, C, G, U\} \tag{7}$$

$$y_i = \begin{cases} 0 \text{ if } x_i = \text{non-m7G sites} \\ 1 \text{ if } x_i = \text{m7G sites} \end{cases} \tag{8}$$

$$z_i = lstm(x_i) \tag{9}$$

$$sigmoid(z_i) = \frac{1}{1 + e^{-y}} \tag{10}$$

### 2.6. Evaluation Parameters

In the literature [15,52–63], the following four equations were employed to measure the prediction performance of the computational method: specificity (sp), sensitivity (sen), accuracy (acc), MCC (Matthew's correlation coefficient), and auROC. In the below equations 'FP' is a false positive, 'TP' is a true positive, 'TN' is a true negative, and 'FN' is a false negative.

$$\begin{cases} Sp = \frac{TN}{TN+FP} \times 100 \\ Sen = \frac{TP}{TP+FN} \times 100 \\ Acc = \frac{TN+TP}{FN+TP+TN+FP} \times 100 \\ MCC = \frac{TP \times TN - FP \times FN}{\sqrt{(TP+FP)(TP+FN)(TN+FP)(TN+FN)}} \times 100 \end{cases} \tag{11}$$

Accuracy: assesses the precision of a computational algorithm for distinguishing m7G sites and non-m7G sites. Sensitivity and specificity are the true positive (TP) and true negative (TN) rates of a test. MCC reveals the correlation between target classes in the case of the imbalanced dataset; here, the ratio of both classes is the same. The area under the ROC curve (auROC) is another measurement metric that shows the predicted outcomes of the model. The auROC indicates the quality of the model. In the above equation, FN and FP denote false negative and false positive, respectively.

### 3. Results and Discussion

An intelligent computational method, namely m7G-LSTM, is designed based on a natural language processing approach, i.e., word2vec, in combination with the deep learning algorithm LSTM. The efficiency is reported on the basis of various measuring metrics, which are mentioned above. The proposed m7G-LSTM model has an accuracy of 95.95%, specificity of 95.97%, the sensitivity of 95.94%, MCC of 0.919, and auROC of 0.980 on the LSTM model, whereas the CNN model achieved 94.94% accuracy, 93.28% specificity, 96.62% sensitivity, 0.899 MCC, and 0.979 auROC. Table 2 shows the detailed projected outcomes of the proposed predictor on LSTM and CNN.

**Table 2.** Performance of CNN and LSTM models using cross-validation.

| Models | Sensitivity | Specificity | Accuracy | MCC |
|--------|-------------|-------------|----------|-----|
| LSTM | 95.94 | 95.97 | 95.95 | 0.919 |
| CNN | 96.62 | 93.28 | 94.94 | 0.899 |

Figures 3 and 4 show the auROC as well as a graphical illustration of the confusion matrix.

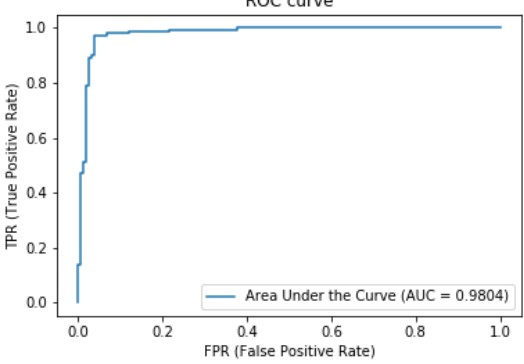

**Figure 3.** The auROC curve of the m7G-LSTM model.

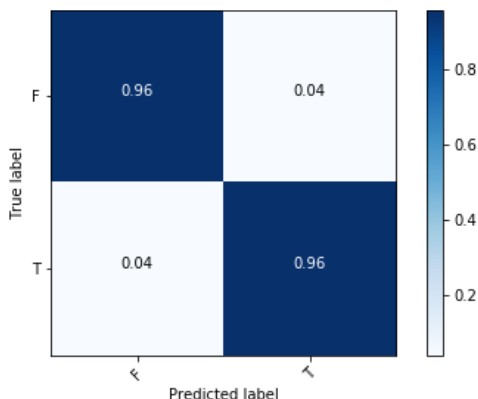

**Figure 4.** Confusion matrix of the proposed m7G-LSTM model.

We compared our proposed m7G-LSTM model to the state-of-the-art models, such as m7GFinder, m7G-IFL, and iRNA-m7G, to calculate its predictive performance. In this study, we develop two various deep learning-based approaches i.e., LSTM and CNN approach to present the proposed m7G-LSTM model. As a result, we compared the highest-performing model of the m7G-LSTM to the most recent m7G-IFL model. To achieve a valid comparison, we execute and evaluate the proposed model on the same benchmark dataset as existing models. Table 3 summarizes the prediction performance. Our computational model, as can be shown, outperforms the other three models, with an accuracy of 95.95%, the sensitivity of 95.94%, specificity of 95.97%, and MCC of 0.919, respectively. Our proposed m7G-LSTM computational model outperformed the existing latest m7G-IFL predictor by 3.54% in specificity, 3.37% in sensitivity, 3.45% in accuracy, and 0.069 in MCC. We observe that our predictor outperforms other models, with an AUROC of 0.980. Our m7G-LSTM model improves upon state-of-the-art prediction models for predicting m7G site modification, based on the results.

**Table 3.** Model comparison between the proposed m7G-LSTM and current models.

| Models | Accuracy | Sensitivity | Specificity | MCC |
| --- | --- | --- | --- | --- |
| m7G-LSTM | 95.95 | 95.94 | 95.97 | 0.919 |
| m7G-IFL [18] | 92.5 | 92.4 | 92.6 | 0.850 |
| m7GFinder [14] | 89.9 | 90.8 | 89.1 | 0.799 |
| iRNA-m7G [15] | 89.9 | 89.1 | 90.7 | 0.798 |
| m7GPredictor [9] | 85.70 | 83.2 | 88.2 | 0.715 |

The graphical depiction of the execution outcomes is demonstrated in Figure 5 in which the m7G-LSTM method achieved remarkable results compared to the current prediction models. This shows the significance of our proposed model.

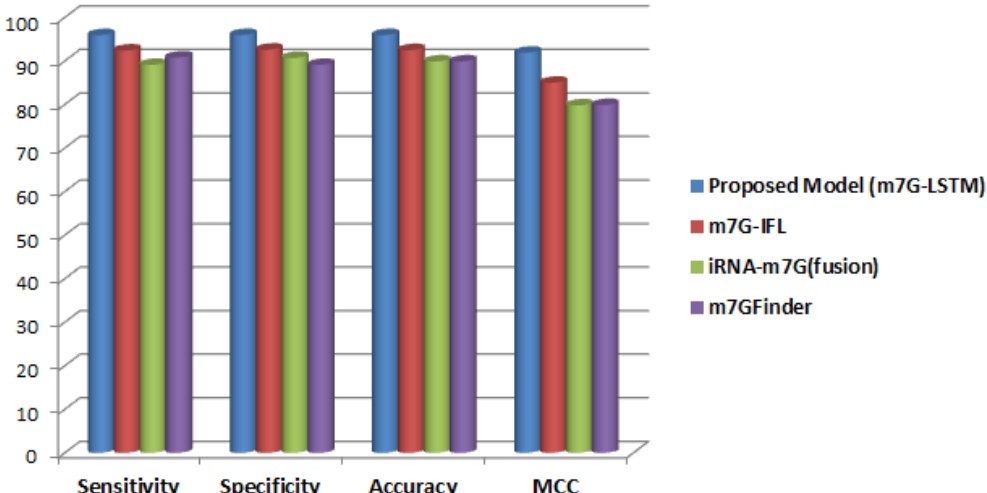

**Figure 5.** Comparison of proposed m7G-LSTM computational model with existing methods.

## 4. Conclusions

The proposed m7G-LSTM method is a reliable and novel deep learning-based prediction model for m7G sites. The proposed model utilized the distributed feature representations that are exploited by the LSTM model. Two-feature encoding schemes were used, i.e., word2vec and one-hot encoding. The input of RNA sequence is divided into 3-mers or words in feature representation, and each word is mapped to its corresponding feature representation using the NLP method, i.e., word2vec. The one-hot encoding converts categorical data to binary data that can be processed by computational models efficiently. Then, the LSTM and CNN models were applied to identify N7-methylguanosine sites, but the LSTM model produced better performance than the CNN model. In terms of all performance measures for discriminating N7-methylguanosine sites, the m7G-LSTM model highly outperforms state-of-the-art models, according to the prediction results. The predicted outcome demonstrates that the proposed m7G-LSTM computational system is efficient and reliable and that it might be useful in drug-related applications and academics.

**Author Contributions:** Conceptualization, M.T.; methodology, M.T.; software, M.H. and R.K.; validation, M.T., M.H., R.K. and K.T.C.; resources, K.T.C.; writing—original draft, M.T. and R.K.; writing—review & editing, M.T., M.H. and K.T.C.; visualization, M.T. and R.K.; supervision, M.H. and K.T.C.; project administration, K.T.C.; funding acquisition, K.T.C. All authors have read and agreed to the published version of the manuscript.

**Funding:** This work was supported by the National Research Foundation of Korea (NRF) grant funded by the Korean government (MSIT) (No. 2020R1A2C2005612).

**Conflicts of Interest:** The authors declare no conflict of interest.

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
