# Peer review of "An Effective Deep Learning-Based Architecture for Prediction of N7-Methylguanosine Sites in Health Systems"

_electronics, doi:10.3390/electronics11121917_

Round 1

Reviewer 1 Report

The authors must improve the literature review part and some extensive supervised based approaches can be added in order to make draft compatible. I advise authors to read some interesting papers and see how they have incorporated literature along with the problem.

For prediction problems:

  1. Saini, Vikash Kumar, et al. "Short term forecasting based on hourly wind speed data using deep learning algorithms." 2020 3rd International Conference on Emerging Technologies in Computer Engineering: Machine Learning and Internet of Things (ICETCE). IEEE, 2020.
  2. Sharma, A. K., Saxena, A., Soni, B. P., & Gupta, V. (2018, March). Voltage stability assessment using artificial neural network. In 2018 IEEMA Engineer Infinite Conference (eTechNxT)(pp. 1-5). IEEE.
  3. Saxena, A. (2021). Grey forecasting models based on internal optimization for Novel Corona virus (COVID-19). Applied Soft Computing111, 107735.

Author are advised to conduct a strong editorial check and quality of figure must be improved. Some additional discussion should be added in results and simulation parts.

Author Response

The authors must improve the literature review part and some extensive supervised based approaches can be added in order to make draft compatible. I advise authors to read some interesting papers and see how they have incorporated literature along with the problem.

For prediction problems:

  1. Saini, Vikash Kumar, et al. "Short term forecasting based on hourly wind speed data using deep learning algorithms." 2020 3rd International Conference on Emerging Technologies in Computer Engineering: Machine Learning and Internet of Things (ICETCE). IEEE, 2020.
  2. Sharma, A. K., Saxena, A., Soni, B. P., & Gupta, V. (2018, March). Voltage stability assessment using artificial neural network. In 2018 IEEMA Engineer Infinite Conference (eTechNxT)(pp. 1-5). IEEE.
  3. Saxena, A. (2021). Grey forecasting models based on internal optimization for Novel Corona virus (COVID-19). Applied Soft Computing111, 107735.

Answer: Thanks to the reviewer for raising this point. The mentioned references are cited in revised manuscript in order to strengthen the literature.

Reviewer 2 Report

The paper does not adequately present the related work. che nel settore e molto ampio. In particular, some of the works in the literature appear to be inspirational of this work and are not even mentioned.

The most important part of the work (i.e., the results) is discussed in an exhaustive way. Results should be compared with similar studies

Author Response

The paper does not adequately present the related work. che nel settore e molto ampio. In particular, some of the works in the literature appear to be inspirational of this work and are not even mentioned.

The most important part of the work (i.e., the results) is discussed in an exhaustive way. Results should be compared with similar studies

Answer: Thanks to the reviewer for pinpointing issue in related work. Now, the paper is revised and rephrased all the ambiguous sentences and made presentable the related work. The results are now compared with existing studies and the discussion section is made precise and concise.

Reviewer 3 Report

-lines 18-19 the acronyms should be explained (m7G-LSTM) together with mentioning it 

-line 23 (specify what is MCC)

-line 45: "In a study" ->  susbstitute with in [Num_OF_Reference]

-line 47 remove comma before [4]

-I would suggest the authors also to add more recent papers to the background literature review:

1) Zago, Elisa, et al. "Early downregulation of hsa-miR-144-3p in serum from drug-naïve Parkinson’s disease patients." Scientific reports 12.1 (2022): 1-13.

2)Ning, Qiao, and Mingyu Sheng. "m7G-DLSTM: Intergrating directional Double-LSTM and fully connected network for RNA N7-methlguanosine sites prediction in human." Chemometrics and Intelligent Laboratory Systems 217 (2021): 104398

3)Zhao, Yongchao, et al. "m7G methyltransferase METTL1 promotes post-ischemic angiogenesis via promoting VEGFA mRNA translation." Frontiers in cell and developmental biology 9 (2021): 1376

4)Wang, Linyu, et al. "A novel end-to-end method to predict RNA secondary structure profile based on bidirectional LSTM and residual neural network." BMC bioinformatics 22.1 (2021): 1-15

-please rephrase the end of the introduction with more details on the LSTM architecture and the pipeline of the work proposed 

-line 101: the mathematical description of the dataset is not sufficient. Please rephrase, explain the variables present and describe 

-2.2 section needs enlarging, more description. Also change the title which is too general 

-2.3 also this section title should be changed. Moroever more examples should be given of the word2vec conversion and the applicability to RNA 

-Eq. 2-3-4-5-6 needs explanation and descriptions of all of the variables. Please specify and describe 

-2.5 the section needs improvement. RNN is not another type of deep learning. Please be careful, re-read and improve the whole section 

-Figure 1: gives further description in the caption and in the text 

-Equations 7-8-9: give more detail, describe all of the variables and notation introduced

-182-183: Even if briefly define what TN, TP and so on are. Also this section needs rephraseing and re-writing 

-Table 2 performances: Are those the result of a cross validation? How was the dataset divided? Was cross validation performed? 

-Figure 3: This confusion matrix is referring to which experiment? Is it referring to the test performances? 

-Figure 4: the comparison is made on the same test set? 

-Conclusions should be enlarged. More descriptions should be given 

Minor Comment: Check the font used, as it is not coherent between sections, tables, figures and text 

Author Response

Comments and Suggestions for Authors

-lines 18-19 the acronyms should be explained (m7G-LSTM) together with mentioning it 

-line 23 (specify what is MCC)

-line 45: "In a study" ->  susbstitute with in [Num_OF_Reference]

-line 47 remove comma before [4]

Answer: Thanks to the reviewer for highlighting the above mentioned phrases. Now, it is corrected accordingly.

-I would suggest the authors also to add more recent papers to the background literature review:

1) Zago, Elisa, et al. "Early downregulation of hsa-miR-144-3p in serum from drug-naïve Parkinson’s disease patients." Scientific reports 12.1 (2022): 1-13.

2)Ning, Qiao, and Mingyu Sheng. "m7G-DLSTM: Intergrating directional Double-LSTM and fully connected network for RNA N7-methlguanosine sites prediction in human." Chemometrics and Intelligent Laboratory Systems 217 (2021): 104398

3)Zhao, Yongchao, et al. "m7G methyltransferase METTL1 promotes post-ischemic angiogenesis via promoting VEGFA mRNA translation." Frontiers in cell and developmental biology 9 (2021): 1376

4)Wang, Linyu, et al. "A novel end-to-end method to predict RNA secondary structure profile based on bidirectional LSTM and residual neural network." BMC bioinformatics 22.1 (2021): 1-15

Answer: Thanks to the reviewer. Now the mentioned studies are cited in the revised manuscript in order to strengthen the presentation of the paper.

-please rephrase the end of the introduction with more details on the LSTM architecture and the pipeline

of the work proposed 

Answer: Thank to the reviewer. Now the end of introduction section is rephrased. Kindly see the end of introduction section in the revised manuscript.

-line 101: the mathematical description of the dataset is not sufficient. Please rephrase, explain the variables present and describe 

-2.2 section needs enlarging, more description. Also change the title which is too general 

-2.3 also this section title should be changed. Moroever more examples should be given of the word2vec conversion and the applicability to RNA 

-Eq. 2-3-4-5-6 needs explanation and descriptions of all of the variables. Please specify and describe 

-2.5 the section needs improvement. RNN is not another type of deep learning. Please be careful, re-read and improve the whole section 

-Figure 1: gives further description in the caption and in the text 

-Equations 7-8-9: give more detail, describe all of the variables and notation introduced

-182-183: Even if briefly define what TN, TP and so on are. Also this section needs rephraseing and re-writing 

Answer: Thanks to the reviwer for highlighting the above mentioned issues. Now all the above mentioned issues are addressed in revised manuscript. Kindly see the revised manuscript.

-Table 2 performances: Are those the result of a cross validation? How was the dataset divided? Was cross validation performed? 

Answer: Thanks to the reviewer. The results reported in Table 2 are cross validation. However, the dataset is divided into three parts 70% for training, 10% for validation and 20% for testing. It is also mentioned the revised manuscript.

-Figure 3: This confusion matrix is referring to which experiment? Is it referring to the test performances? 

Answer: Thanks to the reviewer for pinpoint this issue. Confusion matrix only show how the results are generated. It has four parts, True positive, True negative, False positive, and False Negative. On the basis of these four parts all the measures are computed. It shows the frequency of each elements.

-Figure 4: the comparison is made on the same test set? 

Answer: Thanks to the reviewer. Yes, the comparison was made on same dataset.

-Conclusions should be enlarged. More descriptions should be given 

Answer: Thanks to the reviewer for highlighting is point. Now, the conclusion section is made large and meaningful. Kindly see the revised manuscript.

Minor Comment: Check the font used, as it is not coherent between sections, tables, figures and text 

Answer: Thanks to the reviewer. Now the whole manuscript is checked thoroughly for font, typo, grammatical and other issues. All are resolved. Kindly see the revised manuscript.

Round 2

Reviewer 2 Report

My opinion about the paper is not changed

Author Response

The paper does not adequately present the related work. che nel settore e molto ampio. In particular, some of the works in the literature appear to be inspirational of this work and are not even mentioned.

The most important part of the work (i.e., the results) is discussed in an exhaustive way. Results should be compared with similar studies

Answer: Thanks to the reviewer for pinpointing issue in related work. Now, the paper is revised and rephrased all the ambiguous sentences and made presentable the related work. The results are now compared with some other existing studies and the discussion section is made precise and concise. Some other relevant studies were discussed and cited in introduction section in order to strengthen the introduction section. Kindly see the page no. from 3 to 5 of the introduction section and Table 2 on page no 12 of the revised manuscript.

Reviewer 3 Report

The authors have addressed the comments suggested. 

Some further comments and suggestions: 

-number the mathematical equations (for example the one in section 2.1 is not numbered). Equations should be numbered and referred to

-section 2.2 the encoding should be better described, and defined. In particular the encoded forms example is not clear, and should be described thorughly in a more descriptive way

-still find the equations 2 to equation 6 not clear and in need of more description to be understood better 

-add more description to Figure 7

-Accuracy reported in Table 2 is of cross validation? In case report the specification of this in the text, caption and the rest of the paper 

Author Response

Comments and Suggestions for Authors

-number the mathematical equations (for example the one in section 2.1 is not numbered). Equations should be numbered and referred to

Answer: Thanks to the reviewer for highlighting this point. Kindly see the page no. 5 of the revised manuscript.

-section 2.2 the encoding should be better described, and defined. In particular the encoded forms example is not clear, and should be described thorughly in a more descriptive way

Answer: Thanks to the reviewer for highlighting this point. Kindly see the page no. 6 of the revised manuscript.

-still find the equations 2 to equation 6 not clear and in need of more description to be understood better 

Answer: Thanks to the reviewer for highlighting this point. Kindly see the page no. 8 of the revised manuscript.

-add more description to Figure 7

Answer: Thanks to the reviewer for highlighting this point. There is no figure 7 in manuscript Kindly see the revised manuscript.

-Accuracy reported in Table 2 is of cross validation? In case report the specification of this in the text, caption and the rest of the paper.

Answer: Thanks to the reviewer for highlighting this point. Kindly see the page no. 6 and 11 of the revised manuscript.

Round 3

Reviewer 2 Report

No significant improvements were made to the paper.

Author Response

Reviewer #2: No significant improvement was made to the paper.

Answer: We are very thankful to you for your valuable comments.

(i)        We believe that we have addressed comments of the reviewers in the previous two versions as we have revised the literature review section of the paper by adding more relevant literature, preferably those which are related to the proposed work and problem statement, as highlighted in RED in the revised paper for easy follow up page. [3-5].

However, if the reviewer still believes that his/her comments are not addressed yet, we are happy to do so in future, but we would be grateful if he/she may provide or suggest relevant papers which are needed to be accommodated in the literature reviewer section of the paper.

(ii)       Secondly, we believe that a comparative study of the proposed and existing state of the art approaches were presented, however, as suggested by the valuable reviewer, we have revised the result and discussion section by comparing performance of the proposed scheme against field proven approaches as highlighted in Red for easy follow up page. [12] table 2. However, if the reviewer still believes that his/her comments are not addressed yet, we are happy to do so in future, but we would be grateful if he/she may suggest relevant papers which may be used as a benchmark for comparison.

Reviewer 3 Report

I thank the authors for having addressed my comments

Author Response

Reviewer #3: Thank you for addressing my comments. I have no more comments on this paper and I think it can be published now.

Answer: Thank you very much for your time and valuable suggestions, which have improved the quality of the paper